# The Binding Properties of Antibodies to Z-DNA in the Sera of Normal Healthy Subjects

**DOI:** 10.3390/ijms25052556

**Published:** 2024-02-22

**Authors:** David S. Pisetsky, Matthew J. Gedye, Lawrence A. David, Diane M. Spencer

**Affiliations:** 1Medical Research Service, Durham Veterans Administration Medical Center, Durham, NC 27705, USA; diane.m.spencer@duke.edu; 2Division of Rheumatology and Immunology, Duke University Medical Center, Durham, NC 27710, USA; 3Department of Molecular Genetics and Microbiology, School of Medicine, Duke University, Durham, NC 27710, USA; matthew.gedye@duke.edu (M.J.G.); lawrence.david@duke.edu (L.A.D.); 4Duke Microbiome Center, School of Medicine, Duke University, Durham, NC 27710, USA

**Keywords:** DNA, antibodies, anti-DNA, antinuclear antibodies, B-DNA, Z-DNA, conformation, biofilms, isotype, antigen

## Abstract

Antibodies to DNA are a diverse set of antibodies that bind sites on DNA, a polymeric macromolecule that displays various conformations. In a previous study, we showed that sera of normal healthy subjects (NHS) contain IgG antibodies to Z-DNA, a left-handed helix with a zig-zig backbone. Recent studies have demonstrated the presence of Z-DNA in bacterial biofilms, suggesting a source of this conformation to induce responses. To characterize further antibodies to Z-DNA, we used an ELISA assay with brominated poly(dGdC) as a source of Z-DNA and determined the isotype of these antibodies and their binding properties. Results of these studies indicate that NHS sera contain IgM and IgA as well as IgG anti-Z-DNA antibodies. As shown by the effects of ionic strength in association and dissociation assays, the anti-Z-DNA antibodies bind primarily by electrostatic interactions; this type of binding differs from that of induced anti-Z-DNA antibodies from immunized animals which bind by non-ionic interactions. Furthermore, urea caused dissociation of NHS anti-Z-DNA at molar concentrations much lower than those for the induced antibodies. These studies also showed IgA anti-Z-DNA antibodies in fecal water. Together, these studies demonstrate that antibodies to Z-DNA occur commonly in normal immunity and may arise as a response to Z-DNA of bacterial origin.

## 1. Introduction

Antibodies to DNA (anti-DNA) are a diverse set of antibodies that can bind DNA, a polymeric macromolecule present in both intracellular and extracellular spaces [1,2]. In systemic lupus erythematosus (SLE), a prototypic systemic autoimmune disease, antibodies to double-stranded DNA in the B-DNA conformation (the classic Watson–Crick double helix) are unique markers for classification as well as disease activity [3]. These antibodies can bind to extracellular DNA to form immune complexes which deposit in the kidney to incite nephritis; immune complexes can also stimulate cytokine production by innate immune cells [2,4]. Because SLE anti-DNA antibodies bind to the phosphodiester backbone of B-DNA, they can recognize B-DNA independent of the base sequence [5].

In contrast to patients with SLE, otherwise healthy individuals (normal healthy subjects or NHS) can produce antibodies that bind selectively to determinants on certain bacterial and viral DNA [6,7,8]. These determinants, which are likely dependent on sequence, are present on only some foreign DNA molecules, thereby distinguishing them from B-DNA. The anti-DNA antibodies in NHS sera are not detected in the anti-DNA assays used for routine clinical determinations, however [9]. While preventing confusion in patient evaluation, this lack of activity has led to the erroneous conclusion that anti-DNA expression occurs only in autoimmunity.

As now recognized, DNA can assume a variety of conformations that depend on the base sequence, base modification, and ambient conditions [10]. Among these conformations, Z-DNA is a left-handed helix with a zig-zig orientation, hence the term Z-DNA [11,12]. In contrast, B-DNA is a right-handed helix. Z-DNA occurs preferentially with alternating purine–pyrimidine sequences. The Z-DNA structure is energetically unfavorable at ordinary ambient conditions but the transition from B-DNA to Z-DNA can occur with supercoiling or high concentrations of salt (e.g., 4 M NaCl) [13]. Studies using algorithms to predict Z-DNA structure indicate that Z-DNA may occur at certain sites on chromosomal DNA which “flip in” and “flip out” of the Z-DNA conformation for regulatory interactions [14,15,16].

In structural studies on Z-DNA, chemically modified synthetic DNA compounds such as brominated poly(dGdC) [Br-poly(dGdC)] have been commonly used as a source of Z-DNA since they display the Z-DNA conformation in ordinary salt conditions [17]; this property facilitates the investigation of the immunogenicity and antigenicity of Z-DNA. As these studies have demonstrated, immunization of animals with Z-DNA compounds can induce the production of anti-Z-DNA antibodies under conditions in which B-DNA is essentially inert [18,19,20]. These findings have suggested that Z-DNA, unlike B-DNA, fails to induce tolerance and can be recognized as foreign; in this conceptualization, the failure of Z-DNA to induce tolerance reflects its rarity or transience in chromosomal DNA [21].

Along with their expression of antibodies to B-DNA, patients with SLE commonly express antibodies to Z-DNA [22,23,24,25,26]. While some of these antibodies are specific for Z-DNA, others bind both B-DNA and Z-DNA. The expression of antibodies to Z-DNA differs from that of antibodies to B-DNA since anti-Z-DNA antibodies may occur in other immune-mediated diseases (e.g., rheumatoid arthritis and inflammatory bowel disease) without antibodies to B-DNA [23,27]. The nature of the anti-Z-DNA responses of NHS has been less clear, however, as studies have produced inconsistent findings, including negative results [22,26].

While most studies of Z-DNA have focused on its intracellular role in heredity and gene expression, recent studies provide a new perspective on the presence of Z-DNA in the extracellular space and therefore its potential physiologic (and possibly pathologic) role in that location. Thus, both in vivo and in vitro studies have demonstrated that DNA is an important structural component of bacterial biofilms [28,29,30,31]; biofilms are highly organized structures that enclose and protect bacterial communities as they grow. Furthermore, seminal studies have demonstrated that DNA in biofilms can undergo a B-DNA to Z-DNA transition as the biofilm matures, thus identifying an abundant source of extracellular Z-DNA which could induce anti-Z-DNA antibody production. These studies have also suggested that antibodies to Z-DNA can promote biofilm formation, prompting further investigation into the antibody response to Z-DNA in both healthy subjects and patients with inflammatory and infectious diseases [28].

To elucidate further antibody responses to Z-DNA, in the current study, we have assessed the specificity and immunochemical properties of these antibodies in NHS sera. For this purpose, we have used Br-poly(dGdC) as a source of Z-DNA in an ELISA. As results presented herein demonstrate, NHS sera have significant levels of IgG anti-Z-DNA antibodies whose binding depends on electrostatic interactions. We also show binding activity of IgA and IgM antibodies to Z-DNA in NHS sera as well as IgA antibodies in gastrointestinal secretions. Together with other studies, these findings indicate that Z-DNA is a target of antibodies in both normal and aberrant immunity, with the generation of anti-Z-DNA antibodies perhaps arising in response to extracellular Z-DNA in biofilms from infecting or colonizing microorganisms.

## 2. Results

### 2.1. Isotype Distribution of Antibodies to Z-DNA in Sera

To explore further expression of anti-Z-DNA in NHS sera, we assayed additional NHS sera and extended the analysis to IgM and IgA as well as IgG; the previous study assessed only IgG [24]. In this analysis, we used a standard ELISA format in which DNA preparations are coated onto wells of microtiter plates to serve as antigens. To determine the isotype of bound antibodies, we used horseradish peroxidase conjugated anti-Ig reagents (anti-IgG, anti-IgM or anti-IgA). These reagents had been previously titered to provide similar levels of detection.

In these experiments, we assayed antibodies to different DNA sources using calf thymus (CT) as a source of mammalian DNA in the B-DNA conformation; *E. coli* (EC) DNA as a bacterial DNA which has both B-DNA and non-conserved bacterial DNA sequences; and *Micrococcus luteus* (MC) DNA which has B-DNA, non-conserved bacterial DNA sequences and Z-DNA determinants. Uncoated wells and wells coated with poly(dGdC) served as controls for background. As data in Figure 1 indicate, the NHS sera showed significant IgG binding to Br-poly(dGdC) and MC DNA, with less binding to EC DNA; binding to CT DNA was similar to that of the background. The sera showed IgM and IgA binding to Br-poly(dGdC) and MC DNA. Among sera, the relative amount of IgA and IgM binding to Br-poly(dGdC) was greater than that to MC DNA, whereas IgG responses to MC DNA were greater than those to Br-poly(dGdC). The lack of significant binding by IgM indicates that the antibodies to Z-DNA are unlikely to be so-called natural autoantibodies.

### 2.2. The Role of Electrostatic Interactions in Anti-Z-DNA Binding

The next set of experiments used Br-poly(dGdC) as a source of Z-DNA antigen and addressed the role of ionic interactions in IgG anti-Z-DNA in NHS, determining similarity to induced anti-Z-DNA, on the one hand, and SLE anti-Z-DNA, on the other hand [24]. For this purpose, we first assessed the effects of ionic strength on the binding of the monoclonal and polyclonal anti-Z-DNA antibodies [24].

Confirming results of the previous study, increasing ionic strength had very limited effects on either the association or dissociation of either the monoclonal or polyclonal anti-Z-DNA antibodies (Figure 2). As shown in the figure, NHS anti-Z-DNA binding to Br-poly(dGdC) differed from that of monoclonal and polyclonal anti-Z-DNA preparations since the binding of the NHS sera was inhibited by increasing ionic strength. In association assays, inhibition was observed at 500 mM, with some possible increases at 1000 and 2000 mM. In dissociation assays, the 500 and 1000 mM concentrations produced similar decreases; although, the effects dissociation were possibly less than effects on association. This difference may relate to the time course in which DNA binding occurs as observed previously [32]. The effects of increasing ionic strength are notable in view of past results suggesting that antibodies to Z-DNA in SLE result from charge–charge interactions [22,24]. In the effects of ionic strength, the NHS anti-Z-DNA resembled anti-Z-DNA from SLE sera with the notable difference in the lack of cross-reactive binding to B-DNA.

As another measure of binding activity, we studied the effects of urea on antibody dissociation (Figure 3). Consistent with avid antibody binding, urea caused limited dissociation of monoclonal and polyclonal anti-Z-DNA antibodies from Br-poly(dGdC) in concentrations as high as 4 M. In contrast, the dissociation of NHS antibodies from Br-poly(dGdC) occurred with urea at 1, 2 and 4 M, indicating a less avid population.

### 2.3. The Presence of IgA Anti-Z-DNA in Intestinal Secretions

In view of the presence of IgA antibodies in sera, we next evaluated isotype-specific responses in gastrointestinal secretions (Figure 4). For this purpose, we used preparations of fecal water to test antibody levels in the ELISA. As representative data from three of the samples indicate, the fecal water samples showed IgA antibodies in all samples tested. Levels of IgG and IgM were both low. Of note, we observed IgA antibodies to *E. coli* DNA in three of the samples tested. Together, these results extend responses to Z-DNA to antibodies from the gastrointestinal tract.

## 3. Discussion

The results presented herein provide new insights into the antigenicity of DNA and indicate that sera of otherwise healthy individuals display antibodies to the Z-DNA conformation. While prior studies demonstrated anti-Z-DNA responses in patients with SLE and other diseases, they did not consistently find antibodies to Z-DNA in NHS [22]. One study suggested that antibodies to Z-DNA in NHS reflected cross-reactive binding to charged determinants [25]; our previous study showed anti-Z-DNA responses in NHS, albeit at levels lower than those in SLE [24]. Reflecting emerging evidence on the prominence of extracellular Z-DNA in biofilms, we have reinvestigated the expression of anti-Z-DNA in otherwise healthy individuals. With an ELISA using brominated poly(dGdC) as the test antigen, our data provide clear evidence that NHS sera have antibodies to Z-DNA; our data also indicate the presence of IgA anti-Z-DNA antibodies in gastrointestinal secretions as well as blood.

Structurally, Z-DNA differs significantly from B-DNA in the orientation of the helix, the topology of the phosphodiester backbone and the content in chromosomal DNA. Importantly, Z-DNA is an energetically unfavorable conformation, with the transition from B-DNA to Z-DNA requiring factors such as supercoiling and high concentrations of monovalent and bivalent cations; binding proteins may also promote the transition [11,12,13]. Intracellularly, Z-DNA may occur at least transiently with transcription as the RNA polymerase travels down the DNA molecule; certain sequences may also flip in and out of Z-DNA to provide signals for the regulation of gene expression [33].

Unlike B-DNA, Z-DNA is a rare structure in chromosomal DNA and immunologically may therefore differ from B-DNA in establishing and maintaining tolerance. Indeed, Z-DNA can elicit the production of specific antibodies under conditions in which B-DNA is inactive [18,19,20,34]. These immunizations have used as immunogens chemically modified synthetic DNA polymers [e.g., Br-poly(dGdC)] which stably express Z-DNA. The induced antibodies are quite specific for Z-DNA; although, they can vary in fine specificity, reacting with either exposed bases along the backbone or different regions of Z-DNA structure [35]. In general, the induced antibodies do not bind significantly to natural double stranded mammalian DNA, consistent with the rarity of Z-DNA in chromosomal DNA [36].

The anti-Z-DNA responses of patients with SLE have been of particular interest since sera of patients contain antibodies to both B-DNA and Z-DNA. Animal models of SLE also show responses to both B-DNA and Z-DNA [37,38]; preliminary experiments suggest that monoclonal anti-DNA antibodies from autoimmune mice also show cross-reactive binding to B-DNA and Z-DNA (Spencer, Marion and Pisetsky, preliminary results). These responses are consistent with a breakdown in tolerance to B-DNA in SLE but the reason for the production of antibodies to Z-DNA has been more obscure. While a lack of tolerance to Z-DNA may occur because of its transience and rarity, for induction of anti-Z-DNA, a source of Z-DNA would nevertheless appear necessary.

Based on prior studies on anti-Z-DNA in SLE sera and the current findings on anti-Z-DNA in NHS, at least two explanations can account for an anti-Z-DNA response in SLE. The first explanation is that antibodies to Z-DNA arise as a cross-reaction to B-DNA, with the same antibodies recognizing both B-DNA and Z-DNA despite their structural differences. Since antibodies to both B-DNA and Z-DNA depend on electrostatic interactions as we have shown, SLE anti-DNA antibodies stimulated by B-DNA may bind to negative charges on the phosphodiester backbone of both B-DNA and Z-DNA despite differences in the distribution of charge.

The second explanation for the presence of anti-Z-DNA in SLE is an antigen-driven response to foreign DNA that contains Z-DNA with at least some of these antibodies showing cross-reactive binding to B-DNA. Whether anti-DNA antibodies in SLE are induced by B-DNA or Z-DNA, the notable feature is cross-reactivity with both DNA forms. The propensity to produce cross-reactive antibodies in SLE may reflect abnormalities in the B cell repertoire that allow retention of autoreactive B cell precursors [39,40,41]. These abnormalities could lead to a very different pattern of antigen specificity than that observed with immunization of animals. Evidence for this possibility comes from studies of antibody responses to protein autoantigens [42].

While chromosomal DNA may have only limited amounts of Z-DNA, recent studies have provided strong evidence that Z nucleic acids (both DNA and RNA) are present in infection and can stimulate immune responses to both viruses and bacteria; in these responses, ZBP1 (Z binding protein 1) may provide a sensor for Z nucleic acids [43,44,45]. In addition to a role of Z-DNA in intracellular infection, Z-DNA is a key component of bacterial biofilms, with extracellular DNA in the biofilm matrix undergoing a B- to Z-DNA transition as the biofilm matures [28]. Since Z-DNA is resistant to nuclease digestion [17], the formation of biofilms provides a potentially abundant source of Z-DNA to stimulate antibody production. In this regard, we have provided evidence that DNA from certain bacterial sources (e.g., *Micrococcus luteus* and *Mycobacterium tuberculosis*) stably express Z-DNA, representing another foreign source of Z-DNA [36].

In view of findings of Z-DNA in biofilms and potential exposures of otherwise healthy individuals to bacterial or viral DNA with Z-DNA determinants, an antibody response to Z-DNA in otherwise healthy individuals would not be unexpected. Nevertheless, prior studies have produced variable results concerning anti-Z-DNA in NHS [22,26]. Our recent findings help resolve this conundrum. Thus, the current study as well as prior studies by Spencer et al. clearly indicate that NHS sera contain IgG antibodies to Z-DNA [24]. We also have provided evidence indicating the presence of IgA and IgM anti-Z-DNA responses as well as IgA anti-Z-DNA in gastrointestinal secretion. Since bacteria in the microbiome can form biofilms, the IgA response may arise locally in response to Z-DNA in the gut [31]. Of note, we found evidence of IgA antibodies to *E. coli* DNA, suggesting either cross-reactivity with Z-DNA or the presence of antibodies that bind to a non-B-DNA, non-Z-DNA site on bacterial DNA. These findings are under further investigation.

## 4. Materials and Methods

Biological samples. Sera of NHS were obtained from Plasma Services Group (Moorestown, NJ, USA) and Innovative Research (Novi, MI, USA). Additional samples were provided by Dr. Brice Weinberg of Duke University Medical Center.

To prepare fecal water, participants self-collected stool using provided sampling kits along with an insulated bag and an ice pack to enable cold transport of samples from home freezers to the laboratory. At the laboratory, samples were placed in a locked −20 °C drop-off freezer until transfer to a −80 °C freezer for storage until use. Aliquots were made from homogenized stool in mGAM (Gifu anaerobic medium; HiMedia, Kennett Square, PA, USA) using a stomacher and filtration bag (Nasco Whirl-Pak, Pleasant Prairie, WI, USA). Then, stool aliquots were mixed 50:50 with a 50% solution of glycerol–phosphate-buffered saline (PBS) and frozen at −80 °C for later use.

A total of six samples were tested. Because of background issues and inconsistency among duplicates in the first set of experiments involving three samples, the handling of samples was modified to reduce these problems by varying the centrifugation speed and sample dispersal. Results presented come from the modified method of sample handling.

Antibodies. The monoclonal anti-DNA (Z22) and the polyclonal sheep anti-Z-DNA antibodies were purchased from Absolute Antibody (Boston, MA, USA) and Abcam (Cambridge, MA, USA), respectively. For experiments to investigate the isotype of antibodies to Z-DNA and other DNA antigens, peroxidase-conjugated antibodies recognizing human IgG, IgM and IgA were purchased from Sigma-Aldrich (St. Louis, MO, USA).

Antigens. DNA from calf thymus, *E. coli* and *Micrococcus luteus* were all purchased from Sigma-Aldrich. Poly(dGdC) was also purchased from Sigma-Aldrich. Poly(dGdC) was brominated as described with the conversion to Z-DNA verified by the ratio of OD readings at 260 and 295 [17,24]. A 295/260 OD ratio of 0.34 has been shown to be indicative of Z-DNA conformation. Samples were prepared fresh for each experiment; only those with a 295/260 OD ratio of equal or greater than 0.40 were used.

ELISA. Assays for antibodies to DNA were performed using published methods [24,36]. Briefly, poly(dGdC) is brominated to form Br-poly(dGdC), a DNA molecule in the Z-DNA conformation. Br-poly(dGdC) and other DNA antigens were diluted to 2 μg/mL in 1× SSC (Invitrogen-ThermoFisher, Waltham, MA, USA; 150 mM NaCl, 15 mM Sodium Citrate, pH 7.0). Diluted DNA was dispensed into 96 well plates (Immulon 2HB; Invitrogen-ThermoFisher) at 100 μL/well and incubated overnight at 4 °C. The plates were covered with aluminum foil for all subsequent steps to protect Br-poly(dGdC) from the light.

The next day, the plates were washed 3× with 1× phosphate-buffered saline. The plates were blocked for 2 h at room temperature (RT; 19–23 °C) and washed again. Antibody or plasma diluted in Tris ELISA dilution buffer (T-EDB; 150 mM NaCl, 0.1% BSA, 0.05% Tween 20 in 50 mM Tris, pH 7.4) was dispensed at 100 μL per well and incubated at RT for 1 h. On completion, the wells were washed 3× with 1× PBS and then incubated with 100 μL/well of the appropriate species-specific anti-Ig reagent conjugated to HRP (horseradish peroxidase) for 1 h at RT. The secondary antibodies were diluted in 1× PBS ELISA dilution buffer (P-EDB; 0.1% BSA, 0.05% Tween 20 in 1× PBS, pH 7.4). The wells were then washed again with 1× PBS and then incubated for 30 min at RT with 0.015% 3, 3′, 5, 5′-tetramethylbenzidine dihydrochloride, 0.01% H_2_O_2_ in 0.1 M citrate buffer, pH 4.0 solution. At this point an equal volume of 2 M H_2_SO_4_ was dispensed to all wells, and the absorbance at 450 nm was read using a UVmax multi-plate spectrophotometer (Molecular Devices, San Jose, CA, USA). All chemical reagents and secondary antibodies were purchased from Sigma-Aldrich.

To determine the effects of ionic strength on the association of anti-DNA antibody to DNA antigen binding, plasma/sera and antibody samples were diluted in T-EDB with NaCl levels adjusted to give a concentration of NaCl of 150, 500, 1000 or 2000 mM. These diluted samples were then transferred to the blocked plates as described above.

The effects of urea and ionic strength on antibody dissociation were also investigated. Antibodies and antigens were incubated for 1 h at RT in T-EDB containing 150 mM NaCl. On completion, the T-EDB 150 was discarded and replaced with T-EDB containing various concentrations of NaCl (150 mM, 500 mM and 1000 mM) or urea (1 M, 2 M, 4 M or no Urea). The experiment then proceeded as described above.

All determinations were performed in duplicate, with positive and negative controls. Results of duplicate wells were averaged, and standard deviation were calculated.

## 5. Conclusions

In conclusion, the studies presented herein provide further evidence that otherwise healthy individuals produce antibodies to DNA sequences and conformations which can be recognized as immunologically foreign because of their rarity and transience in chromosomal DNA. In some respects, the anti-Z-DNA antibodies in NHS closely resemble those in SLE in terms of binding properties (the role of electrostatic interactions), indicating that a predominant role of electrostatic interactions does not preclude specificity for Z-DNA compared to B-DNA. Future studies will address the intriguing questions of whether antibodies to Z-DNA have functional properties and target sites in bacterial biofilms which are rich sources of Z-DNA.

## Figures and Tables

**Figure 1 ijms-25-02556-f001:**
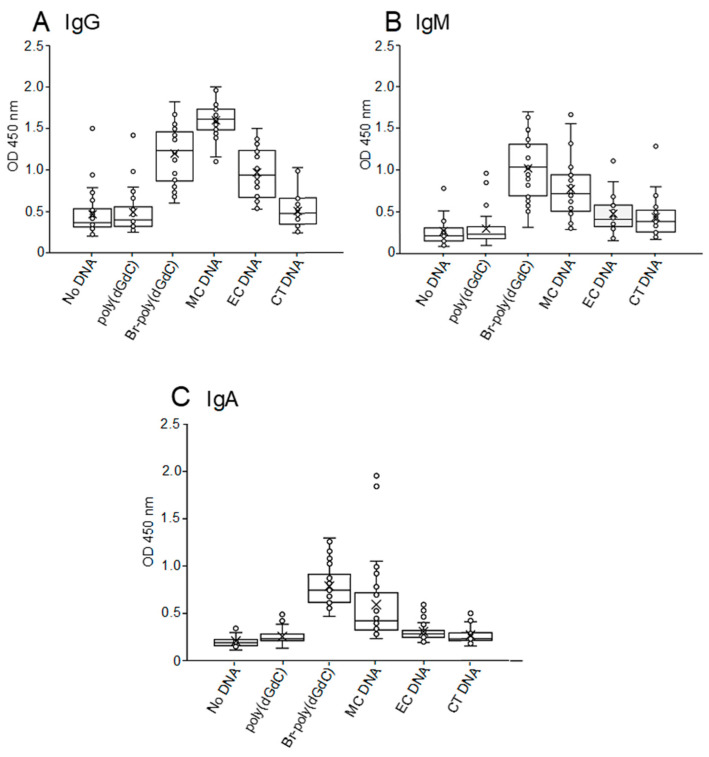
Isotype expression of antibodies to Z-DNA. Plasma and sera from 26 NHS were tested by ELISA at a 1:200 dilution using isotype-specific peroxidase-conjugated anti-Ig reagents for binding to DNA antigens: poly(dGdC), Br-poly(dGdC), *Micrococcus luteus* (MC), *Escherichia coli* (EC) and calf thymus (CT). All determinations were performed in duplicate. The individual 450 nm absorbance values were averaged; the standard deviations are indicated by error bars. Results of determinations of IgG (**A**), IgM (**B**) and IgA (**C**) are shown.

**Figure 2 ijms-25-02556-f002:**
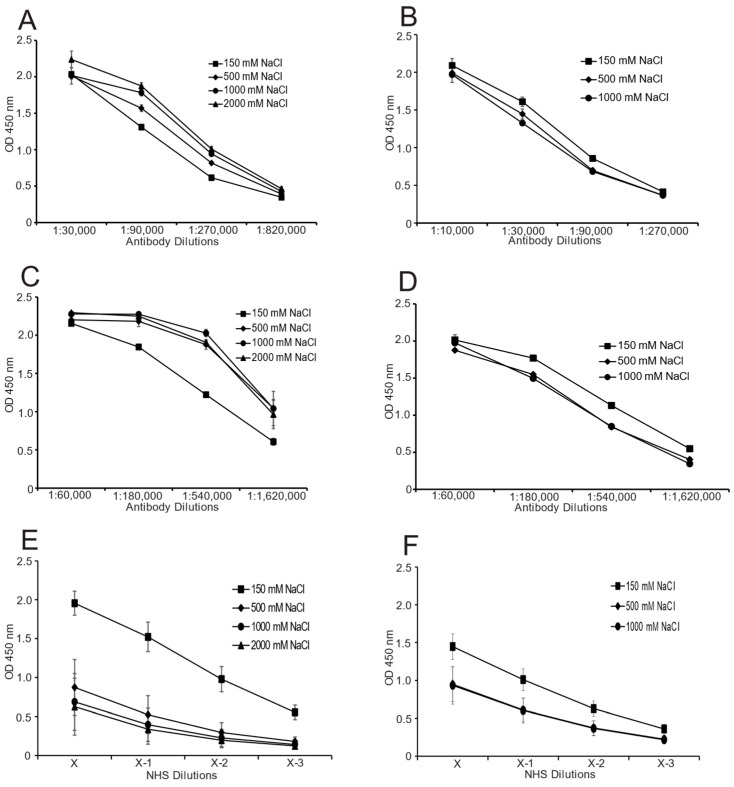
Effects of ionic strength on antibody binding to Z-DNA. The effects of ionic strength on the binding to Z-DNA were tested by ELISA using NHS plasmas/sera, a commercial monoclonal anti-Z-DNA antibody, and a commercial sheep polyclonal anti-Z-DNA preparation in association (**A**,**C**,**E**) and dissociation (**B**,**D**,**F**) and assays. The concentrations of NaCl used were as follows: 150 mM (squares), 500 mM (diamonds), 1000 mM (circles) and 2000 mM (triangles; association only). The binding of commercial anti-Z-DNA antibodies, a sheep polyclonal anti-Z-DNA (**A**,**B**) and a mouse monoclonal anti-Z-DNA (**C**,**D**) were tested under the conditions described. Data in panels (**A**–**D**) were derived from representative experiments. The studies analyzed the binding characteristics of 10 NHS samples for association (**E**) and 12 NHS samples for dissociation (**F**). The NHS samples were diluted in a series of 2-fold dilutions with initial dilutions (X) ranging from 1:100 to 1:300. All assays were performed in duplicate. The 450 nm absorbance values were averaged for the groups; the standard deviations are indicated by error bars.

**Figure 3 ijms-25-02556-f003:**
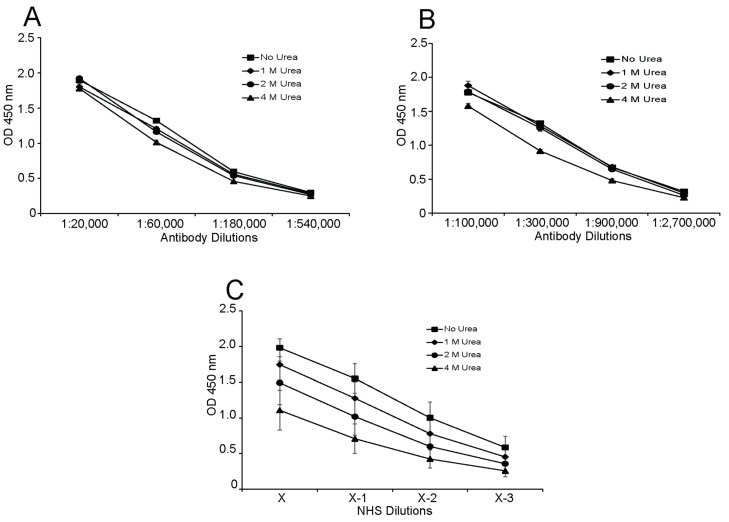
The effects of urea on antibody dissociation. The effects of urea on anti-Z-DNA dissociation were tested as a measure of avidity for (**A**) polyclonal sheep anti-Z-DNA; (**B**) mouse monoclonal anti-Z-DNA; and (**C**) 11 NHS plasmas/sera. The concentrations of urea used were no urea (squares), 1 M (diamonds), 2 M (circles) and 4 M (triangles). The dilutions of polyclonal sheep anti-Z-DNA and mouse monoclonal anti-Z-DNA are indicated. The NHS were diluted in a series of 2-fold dilutions with initial dilutions (X) ranging from 1:100 to 1:200. All assays were performed in duplicate. The 450 nm absorbance values were averaged for the NHS samples; the standard deviations are indicated by error bars.

**Figure 4 ijms-25-02556-f004:**
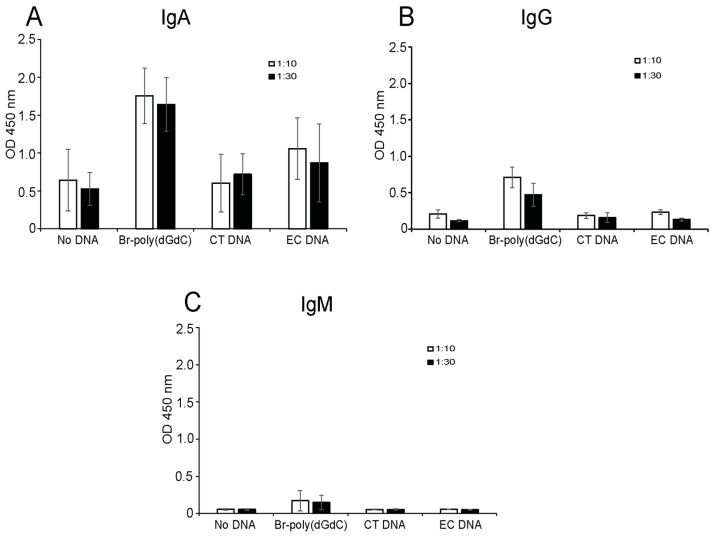
Assay of anti-Z-DNA antibodies in fecal water. Representative data of fecal water samples from three different individuals were analyzed by ELISA at two dilutions [1:10 (unfilled bars) and 1:30 (filled bars) for levels of IgA (**A**), IgG (**B**) and IgM (**C**) levels]. This figure presents averaged data from three samples. Means and standard deviations of the measured OD 450 nm absorbances are shown.

## Data Availability

All data will be made available.

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
