# Peer review of "The Binding Properties of Antibodies to Z-DNA in the Sera of Normal Healthy Subjects"

_ijms, 2024, doi:10.3390/ijms25052556_

Round 1

Reviewer 1 Report

Comments and Suggestions for Authors

This is an interesting study which provides new insights into antigenicity of DNA. The authors use sera from normal healthy subjects and nicely show how the sera of these individuals contain antibodies of IgG, IgM and IgA isotypes targeting Z-DNA. The results are clearly and well presented.

I only have a couple of questions which the authors clarify and discuss.

1) The authors claim that anti-Z-DNA ab are not detected in routine anti-dsDNA assays (CLIFT, ELISA, ALBIA, etc I suppose). But does that exclude the possibility that anti-Z-DNA ab could interfere with the detection of of anti-dsDNA in these assays? Could this be tested?

2) It is clear that the authors used normal healthy blood donor sera but were their any associations with sex or age and levels of anti-Z-DNA?   

Reviewer 2 Report

Comments and Suggestions for Authors

In this brief study, Pisetzky and Spencer demonstrate presence of anti Z-DNA antibodies in sera obtained from normal healthy subjects (NHS). They determine the subclasses of anti Z-DNA antibodies and the potential binding mode using an established ELISA method and commercially available mono- and polyclonal anti Z-DNA antibodies as controls.

They found IgG, IgA and IgM Z-DNA reactive antibodies in these sera. In addition, they found anti Z-DNA IgA in intestinal secretions. They also concluded from their studies that sera antibodies bind Z-DNA by electrostatic interactions, as demonstrated by the salt-sensitivity of binding. This contrasts with the known binding mode of monoclonal antibodies raised against brominated oligo dGdC in the mouse.

The overall main conclusion is that sera of normal healthy individuals - and not only SLE patients -possess anti Z-DNA antibodies, and the authors speculate that this may be the result of Z-DNA in biofilms as antigens in individuals.

Overall, this study is scientifically sound and presented in a clear and concise style. However, there are a few points which the authors may consider in order to increase clarity and impact:

1)    It would add significantly if the ELISA assays could be done individually on the 26 NHS sera. Information on the prevalence of these antibodies is important. For example, it seems possible that only few of the NHSs have these antibodies (at rather high levels). This could just be done with no DNA as control.

2)    A brief introduction in the RESULTS section how the ELISA assay is done would aid the novice to better understand what is measured.

3)    Fig. 2: the data on the salt sensitivity of binding of commercial antibodies should be shown as control.

4)    Fig. 4: formatting is off.

5)    Page 8 bottom: the statement “express Z-DNA” is odd. I guess the authors mean that these genomes contain stretches of DNA in Z-conformation?

6)    Has it been shown with monoclonal SLE antibodies that they bind both B-DNA and Z-DNA?
